# Applicability and safety of discontinuous ADVanced Organ Support (ADVOS) in the treatment of patients with acute-on-chronic liver failure (ACLF) outside of intensive care

**L. Kaps**[1,2,3], **C. J. Ahlbrand**[1,2], **R. Gadban**[1], **M. Nagel**[1,2], **C. Labenz**[1,2], **P. Klimpke**[1],
**S. Holtz**[1], **S. Boedecker**[1], **M. Michel**[1,2], **W. M. Kremer**[1,2], **M. Hilscher**[1,2], **P. R. Galle**[1],
**D. Kraus**[1], **J. M. Schattenberg**[1,4], **J. Weinmann-Menke**[1] *

1 Department of Internal Medicine I, University Medical Centre of the Johannes Gutenberg-University, Mainz,
Germany, 2 Cirrhosis Centre Mainz (CCM), University Medical Centre of the Johannes Gutenberg-
University, Mainz, Germany, 3 Institute of Translational Immunology, University Medical Centre of the
Johannes Gutenberg-University, Mainz, Germany, 4 IMetabolic Liver Research Program; I. Department of
Medicine, University Medical Centre of the Johannes Gutenberg-University, Mainz, Germany

* julia.weinmann-menke@unimedizin-mainz.de

Medicine, JAPAN

**Data Availability Statement:** The data underlying
this study are available on Zenodo (DOI: 10.5281/
zenodo.4624885).

## Abstract

### Background

ADVanced Organ Support (ADVOS) is a novel type of extracorporeal albumin dialysis and
holds promise to sustain liver function and recovery of patients with acute-on-chronic liver
failure (ACLF). Previously, ADVOS was tested as continuous treatment for intensive care
patients with liver failure. Data related to the applicability and safety as discontinuous treat-
ment outside of ICU is not available.

### Aim

Evaluation of ADVOS as discontinuous treatment for patients with ACLF outside intensive
care unit and comparison with a matched historic cohort.

### Methods and results

In this retrospective study, 26 patients with ACLF and the indication for renal replacement
therapy related to HRS-AKI were included. Majority of patients were male (65%) with alco-
holic cirrhosis in 88% and infections as a trigger of ACLF in 96%. Liver function was severely
compromised reflected by high median MELD and CLIF-C ACLF scores of 37 (IQR 32;40)
and 56.5 (IQR 51;60), respectively. Patients were treated discontinuously with ADVOS over
a median time of 12 days (IQR 8.25;17) and received 8 (IQR 4.25;9.75) treatment cycles on
average. No treatment related adverse events were recorded, and safety laboratory param-
eters remained constant during the observation time. After 16 h cumulative dialysis therapy,
ADVOS significantly reduced protein-bound bilirubin (14%), creatinine (11.8%) and blood
urea nitrogen (BUN, 33%). Using a matched cohort with ACLF treated with hemodialysis,
ADVOS achieved a stronger decrease in bilirubin (p = 0.01), while detoxification of water-

**Funding:** The author(s) received no specific funding for this work.

**Competing interests:** The authors have declared that no competing interests exist.

**Abbreviations:** ACLF, Acute-on-chronic liver failure; ADVOS, ADVanced Organ Support; BUN, blood urea nitrogen; CRRT, continuous renal replacement therapy; EASL-CLIF, European Association for the Study of the Liver-Chronic Liver Failure; ECAD, extracorporeal albumin dialysis; ELS, Extracorperal liver support; HAS, human albumin solution; HD, Hemodialysis; HRS-AKI, Hepatorenal syndrome-acute kidney injury (AKI); Prothrombin time-INR, International Normalized Ratio; IQR, interquartil range; LTX, liver transplantation; MELD, model for end-stage liver disease; NASH, non-alcoholic steatohepatitis; Prometheus, fractionated plasma separation; SOFA, Sequential Organ Failure Assessment; SPAD, single pass albumin dialysis; SOP, Standard operating procedure.

soluble catabolites' including creatinine and BUN was comparable. The 28-days mortality in the ADVOS group was 56% (14/26) and was not inferior to predicted survival (predicted median 28-days mortality was 44%, IQR 30; 59).

## Conclusion

Discontinuous ADVOS treatment was safe and effective in patients with ACLF outside intensive care and outperformed hemodialysis in reducing protein-bound metabolites.

## Introduction

Acute-on-chronic liver failure (ACLF) is a life-threatening condition which might occur in patients with liver cirrhosis. According to the European Association for the Study of the Liver-Chronic Liver Failure (EASL-CLIF) Consortium ACLF is characterized by an acute decompensation of the cirrhotic liver and includes organ failure and high short-term mortality. Organ failure manifest as hepatic and extra hepatic abnormalities in ACLF including acute kidney failure (56%), hyperbilirubinaemia (44%) followed by coagulopathy (27%), hepatic encephalopathy (24%), circulatory—(17%) or respiratory constrains (9%) [1]. ACLF arises in up to 40% of the patients with decompensation and is more frequent in young patients [2, 3]. In Europe, the most common triggers for ACLF are ongoing alcohol consumption (75%), viral hepatitis related (10%), while in some patients (~15%) the cause remains cryptogenic [1, 4]. The European Association for the Study of the Liver Chronic Liver Failure (EASL-CLIF) consortium set up the CANONIC (EASL-CLIF ACLF in Cirrhosis) study and developed the CLIF-C ACLF score to stratify patients with ACLF and predict 28-days mortality [5, 6].

The therapeutic options for ACLF are limited as donor organ allocation for liver transplantation is competitive and frequently not an option when e.g. alcohol use is ongoing. Thus, novel therapeutic approaches for ACLF aim to support organ function and recovery to bridge especially acute phases of decompensation are urgently needed. Here, Extracorperal liver support (ELS) is an appealing concept to sustain liver function as bridge-to-transplant or even bridge-to-recovery [7]. Hemodialysis, as a renal replacement therapy, primarily eliminates water-soluble toxins, while protein-bound toxins are retained, increasing the risk of secondary organ failure [8]. In contrast, extracorporeal albumin dialysis (ECAD) closes this gap and removes both water soluble and protein-bound toxins from the blood. Several ECAD concepts were already tested in clinics including single pass albumin dialysis (SPAD), Molecular Adsorbent Recirculating System (MARS) and fractionated plasma separation (Prometheus) [9, 10]. ECADs improved secondary endpoints (hemodynamics and hepatic encephalopathy), while primary endpoints (survival probability) and other endpoints were not met [10–12]. However, outcomes may be biased by confounders and insufficient stratified patient cohorts. Further data of studies, which applied the latest definition of ACLF (e.g. CLIF-C ACLF score) are yet missing.

Previous ECAD systems consumed significant amounts of human albumin solution (HAS), a scarce resource, making the procedure expensive and unattractive. The ADVanced Organ Support (ADVOS) is a novel type of ECAD, facilitating multi-organ support. The optimized detoxification and continuous regeneration of toxin-binding albumin reduces the needed HAS to a minimum. In a tertiary circuit, the regeneration of HAS is achieved by alteration of the pH and temperature, modulating the conformation of the protein and, thus, it's binding capacities to toxins (Fig 1).

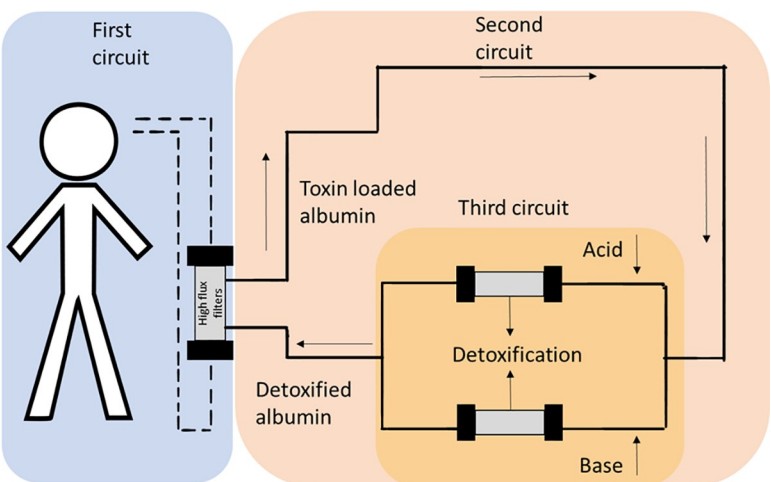

**Fig 1. Schematic representation of ADVOS.**

In preliminary studies, ADVOS proved to be safe and effective as continuous treatment for patients with liver and multi-organ failure in intensive care, but applicability of ADVOS as discontinuous treatment on peripheral ward has not yet been evaluated [7, 13]. The current study explored the applicability and safety of ADVOS as discontinuous treatment in a regular dialysis unit and compared outcome of ADVOS in a case-control study with intermittent hemodialysis (HD) with matched ACLF patients.

## Methods

### Study design

This is a retrospective study evaluating ADVOS as discontinuous treatment for patients with ACLF outside of an intensive care unit. According to clinical standard, patients were treated with ADVOS from July 2018 to November 2020 on a normal ward at the Cirrhosis Center Mainz, University Medical Center Mainz, Germany. Medical history, dialysis parameter and laboratory data were retrieved by chart review or from the local laboratory system, respectively.

### Ethics

The study was conducted according to the ethical guidelines of the 1975 Declaration of Helsinki and its later amendments and in line with the guidelines of the local ethics committee (Ethik-Kommission, Landesärztekammer Rheinland-Pfalz). The study was approved by the local ethics committee. Informed consent was waived due to the retrospective design of the study. Medical data of HD patients was obtained from a local ACLF registry as agreed before by the local ethics committee (reference number: Nr. 837.232.17 [11066]).

### Matching

Detoxification effect of ADVOS vs. regular HD was compared by reduction of the laboratory parameters creatinine, bilirubin and blood urea nitrogen (BUN). Therefore, ADVOS patients were matched 1:1 to ACLF patients who received regular hemodialysis. Matching was done 1:1 based on the following criteria: Sex, age, etiology of cirrhosis, cumulative dialysis time and liver function as reflected by MELD- and CLIF-C ACLF-score. Laboratory data of HD patients

was retrieved from a local ACLF registry, containing 136 patients. Baseline characteristics of patients of the registry are shown in S1 Table.

## Inclusion and exclusion criteria

Patients were included based on the following criteria, while data was retrieved by chart review Table 1:

## ADVOS

The ADVOS system is an ECAD device, facilitating combined liver and kidney support (Fig 1). ADVOS is approved as treatment for patients with reduced detoxification with impaired liver, kidney and lung organ function in Europe. It consists of three communicating circuits. The first circuit circulates the blood over high flux filters (Elisio-19H, Nipro, Ratingen, Germany) back to the patient through conventional double lumen dialysis catheter. In the second circuit, a dialysate containing 2–4% albumin solution runs in parallel to the first circuit only separated by a semipermeable membrane when running through the dialyzer, loading the protein solution with protein-bound toxins and eliminating water-soluble toxins. The third circus represents metaphorically speaking the heart of the ADVOS system, achieving the detoxification of the dialysate and, thus, recycling of the toxin-loaded albumin solution. In this circuit, the dialysate is separated in two branches where either acids or alkalis are substituted. By alteration of the pH, the albumin changes its confirmation, reducing its affinity to the loaded toxins. The effect is further enhanced by lowering the temperature down to ~28˚C. The discharged toxins are removed by a tertiary filtration process. The acidified and alkalized albumin dialysates converge, resulting in a physiological pH after reunion. The temperature of the dialysate is re-warmed up to body temperature. Both parameters, pH and temperature, can be modulated if this is considered useful for therapy. Finally, the regenerated albumin dialysate re-circles to the second circuit for the next cycle. No additional renal replacement therapy, neither serial nor parallel, was performed besides ADVOS. No pre- and post-dilution for ADVOS was conducted.

## Anticoagulation

In line with our clinical standards, anticoagulation was performed by regional anticoagulation wit citrate independent of the patient's prothrombin time.

**Table 1. Inclusion and exclusion criteria.**

| Inclusion | |
|---|---|
| 1. | Clinical or histological evidence of liver cirrhosis[Π] |
| 2. | Acute decompensation indicated by ascites (II-III), deterioration of laboratory parameter or hepatic encephalopathy (west haven criteria grade $\geq$ I) in line with CLIF organ Failure Score $\geq$ 1 |
| 3. | Bilirubin $\geq$ 4 mg/dl and sudden prothrombin time-INR > 2 |
| 4. | HRS-AKI[*] with anuria/oliguria as indicated by creatinine > 2 mg/dl after volume expansion PLUS substitution of albumin for $\geq$48h with subsequent vasopressor $\geq$48 therapy according the EASL HRS/ACLFD guidelines [13] |
| Exclusion | |
| 1. | Any kind of vasopressors and mean arterial pressure $\leq$50 mmHg despite volume expansion |
| 2. | Patients age < 18 years |

[Π]Diagnosis of liver cirrhosis was made by histology, typical appearance in ultrasound or radiological imaging, endoscopic features of portal hypertension, and medical history

[*]HRS-AKI was diagnosed according the latest definition by an experienced nephrologist [14].

## Endpoints

Considering the retrospective character of the study, endpoints of the study were defined as: 1. Applicability of ADVOS as discontinuous treatment regarding patient safety in patients with ACLF in non-intensive care; 2. The effect of ADVOS treatment on standard laboratory parameters after cumulative 16 h of dialysis. This time point was chosen to compare our data to previous reference studies of Huber and Fuhrmann et. al., who have tested ADVOS as continuous treatment for patients with liver and multi-organ failure in intensive care unit [7, 13]; 3. Comparison of detoxification effect of ADVOS vs. HD; 4. Comparison of one-month (28-days) mortality in the matched cohort.

## Statistical analysis

Quantitative data are expressed as medians with interquartile ranges (IQR). Categorical variables are given as frequencies and percentages, respectively. Statistical tests included Mann-Whitney-U-Test or Wilson Cox test for unpaired or paired numeric variables, respectively. Chi-square test was performed for nominal values. Our complete data analysis is exploratory. Log-rank test was performed for survival analysis. Here, the one patient was transplanted due to terminal hepatic failure and was not censored in the analysis, being treated as a complete case (death). Hence, no adjustments for multiple testing were performed. For all tests we used a 0.05 level to define statistically relevant deviations from the respective null hypothesis. However, due to the large number of tests, p-values should be interpreted with caution. Data were analysed using IBM SPSS Statistic Version 25.0 (Armonk, NY: IBM Corp.) and GraphPad Prism Version 8.0.2 (GraphPad Software, California, US).

# Results

In total, 26 patients with ACLF who received ADVOS treatment were retrospectively studied between 06/2018 and 11/2020 (Table 2). 65.3% of patients were male with a median age of 53.5 years (IQR 49; 57.75). Main etiology of cirrhosis was excessive alcohol consumption (88%), while only 3 patients (12%) had a mixed etiology (HBV, alcohol or NASH). Infections (96%) were identified as main trigger for decompensation. All patients were diagnosed with ACLF with a median CLIF-C ACLF score of 56.5 (IQR 51;60), having the indication for renal replacement therapy based on critical laboratory parameters or fluid overload due to HRS-AKI (100%) [5, 14]. Median CLIF Organ Failure Score was 12 (IQR 11;12), while main organ was liver failure 23 (88%) [5]. Patients were severely ill, having a median expected one-month (28-days) mortality of 44% (IQR 30; 59) as calculated by the CLIF-C ACLF score [5]. Since coagulation was compromised reflected by median platelet count of 83 (IQR 60; 132) and median prothrombin time-INR of 2 (IQR 1.7; 2.4), citrate instead of heparin for anticoagulation is routinely used during ADVOS.

## ADVOS treatment

Patients received a median of 8 (IQR 7.25; 9.75) ADVOS cycles as discontinuous treatment over the median period of 12 days (IQR 8.25; 17) on a peripheral ward. Median duration of the first two treatment cycles was 8 h (IQR 7; 8) and was adapted to the specific needs of each patient after the initial treatment cycles. Median blood flow rate was 150 ml/min (IQR 150; 150) and median ultrafiltration rate was 150 ml/h (IQR 110; 250, S2 Table). No adverse events or alarming changes in laboratory parameter were documented (S3 Table).

**Table 2. Baseline characteristics of included patients with ACLF and HRS-AKI.**

| | |
|---|---|
| **Patients, n** | 26 |
| Male, n (%) | 17 (65.3) |
| Age (years), median (IQR) | 53.5 (49; 57.75) |
| **Etiology of cirrhosis, n (%)** | |
| Alcoholic cirrhosis | 23 (88%) |
| Mixed etiology * (HBV, HCV, NASH) | 3 (12%) |
| **Precipitating events as trigger of ACLF** | |
| Infections, n (%) | 25 (96%) |
| Variceal bleeding, n (%) | 1 (4) |
| **Liver function** | |
| Child-Pugh (%) | B (15), C (85) |
| MELD, median (IQR) | 37 (32; 40) |
| **CLIF-C ACLF score** | |
| CLIF-C ACLF score, median (IQR) | 56.5 (51; 60), Grade I: 1, II: 14, III: 11 |
| ACLF grade, median (IQR) | 2 (2; 3) |
| CLIF Organ Failure Score, median (IQR) | 12 (11; 12) |
| Liver failure, n (%) | 23 (88%) |
| Kidney failure[Π], n (%) | 14 (54%) |
| Cerebral failure, n (%) | 0 |
| Coagulation failure, n (%) | 3 (11.5%) |
| Circulatory failure, n (%) | 0 |
| Lung failure, n(%) | 0 |
| One month (28-days) mortality, median % (IQR) | 44 (30; 59) |
| **Laboratory** | |
| Sodium, mmol/l, median (IQR) | 137 (131.5; 139.75) |
| Potassium, mmol/l, median (IQR) | 3.6 (3.4; 4.4) |
| BUN, mmol/l, median (IQR) | 49 (44.3;74) |
| Creatinine, mmol/l, median (IQR) | 4.7 (3.9; 5.3) |
| Prothrombin time-INR, median (IQR) | 2 (1.7; 2.4) |
| Bilirubin, mg/dl, median (IQR) | 23.4 (15.5; 30.75) |
| Thrombocytes, /nl, median (IQR) | 83 (60; 132) |
| **Treatment** | |
| Total ADVOS treatment days, median (IQR) | 12 (8.25;17) |
| Total received ADVOS treatments, median (IQR) | 8 (4.25;9.75) |
| Duration of the first two ADVOS treatments in hours, median (IQR) | 8 (7; 8) |
| Current use of vasopressors | None |

*Viral hepatitis were diagnosed according current guidelines [15, 16]

[Π]diagnosed according the current guideline for HRS-AKI [14].

## Treatment efficacy of ADVOS

The detoxification effect of ADVOS was assessed after the first two treatment cycles (16 h). ADVOS reduced significantly BUN (-16.5, IQR -37.8; -3.5; $p \leq 0.0001$) after cumulative 16 h of dialysis. Beside BUN, bilirubin (-14.5%, IQR 8.3; 29.1) and creatinine (-11.8%, IQR -25.4;4.2) levels were significantly ($p < 0.05$) reduced (Table 3). In a subgroup analysis, we found that the elimination of bilirubin was concentration dependent. Reduction tended to be more efficient at higher bilirubin levels ($\geq 20$ mg/dl) as shown in S4 Table.

**Table 3. Elimination of water- and protein-bound toxins after cumulative 16 h of ADVOS as treatment.**

|  | Before ADVOS | After ADVOS | ΔDelta (mg/dl) | ΔDelta % | p-value |
|---|---|---|---|---|---|
| Serum bilirubin (mg/dL) | 23.4 (15.5;30.75) | 17.1 (11.75;24.5) | -3.4 (-6.7;1.95)* | -14.5 (8.3;29.1)* | 0.034 |
| Serum creatinine (mg/dL) | 4.7 (3.9;5.3) | 3.4 (-6.7;1.95) | -0.6 (-1.2;0.2)* | -11.8 (-25.4;4.2)* | 0.04 |
| BUN (mg/dL) | 49 (44.3;74) | 33.5 (29.3;42.3) | -16.5 (-37.8;-3.5)*** | -33.7 (-7,1;77)*** | 0.00012 |

Median (IQ25, IQ75). Non-parametric paired Wilcoxon test

*p < 0.05

***p < 0.0001. Median treatment duration for one cycle 8 h (7;9).

## Comparison of ADVOS vs. hemodialysis in a case-control study

The detoxification effect of ADVOS was compared to regular hemodialysis (HD) based on the reduction of the laboratory parameters creatinine, bilirubin and BUN in ACLF patients. ADVOS treated patients were matched 1:1 to patients of a local ACLF registry. Beside the need for renal replacement therapy because of HRS-AKI, pairs were found based on sex, age, etiology of cirrhosis, liver function as reflected by MELD score and CLIF-C ACLF score and cumulative dialysis time (Table 4) [17]. In total, 25 matches were found based on the defined criteria, while statistical testing between the matched cohorts revealed no significant difference, indicating successful matching.

For comparison of ADVOS vs. HD, blood parameters were assessed before and after 16 h cumulative dialysis time from day one of renal replacement therapy. ADVOS patients received two treatment cycles with 8 h per day on two consecutive days, while HD patients were treated for 4 h per day on four consecutive days, summing up to 16 h dialysis time for both modalities. Reduction of bilirubin was higher in the ADVOS group compared to patients treated with regular HD (median reduction -2.48 vs. 1.01 mg/dl, p = 0.01). In contrast, no difference was found for creatinine and BUN. (Fig 2).

## 28-days mortality of patients treated with ADVOS vs intermittent hemodialysis

28-days mortality of the matched cohort was analyzed between ADVOS vs. HD treated patients. Despite both cohorts had comparable grade of liver failure (MELD- and CLIF-C

**Table 4. Matching of ADVOS vs HD treated patients (p-values above cut-off (>0.05) indicate no significant difference between the matched cohorts).**

| Matching criteria | Variables | ADVOS n = 25 | HD n = 25 | p-value |
|---|---|---|---|---|
| 1 | Sex | 25 of 25 (8 vs. 8 females, 17 vs. 17 males) patients | | |
| 2 | Age, median (IQR) | 53.5 (49; 57.8) | 59 (52; 63) | 0.082* |
| 3 | Same Etiology of cirrhosis | 22 of 25 (85%, 22 vs. 22 alcoholic cirrhosis and 3 alcoholic cirrhosis vs. 3 mixed etiology cirrhosis) patients | | 0.43‡ |
| 4 | HRS-AKI Π | 25 of 25 (100%) patients | | |
| 5 | MELD score, median (IQR) | 37 (32.3; 40) | 34 (31; 39) | 0.2* |
| 6 | CLIF-C ACLF score*, median (IQR) | 56.6 (51; 60) | 65 (54.75; 73) | 0.28* |
| 7 | One month (28-days) mortality, median % (IQR) | 44 (30; 59) | 47 (29; 60) | 0.85* |
| 8 | Duration of Dialysis | 16 h (100%) | | |

‡Chi-squared test

*Mann-Whitney-U-Test

Πdiagnosed according the current guideline for HRS-AKI[14].

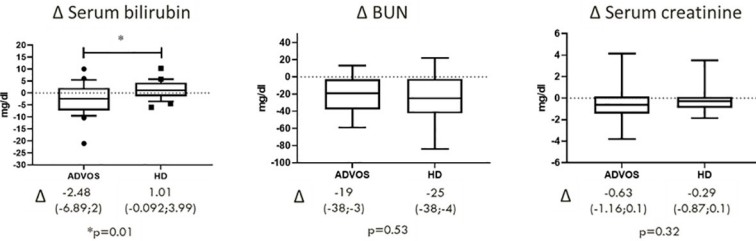

**Fig 2. Detoxification effect of ADVOS vs. HD after cumulative 16 h dialysis time in the matched cohort.** ADVOS outperformed HD for the reduction bilirubin, while detoxication was similar for BUN and creatinine. (Median (IQR) of delta Δ blood parameters before and after dialysis; p-values were calculated by Wilson-cox test).

ACLF score) and baseline characteristics (Table 4), in the ADVOS treated cohort 11 of 25 (56%) patients died, while 15 of 25 (40%) patients in the HD treated cohort did not survived for 28-days mortality (Fig 3). In addition, predicted survival by the CLIF-C ACLF score was 44% (IQR 30; 59, Table 2) in the ADVOS cohort, suggesting that ADVOS does not negatively impact survival. However, there was no substantial difference of long-term survival between ADVOS vs. HD treated patients, while only one patient in the ADVOS cohort received liver transplantation (S1 Fig).

## Discussion

ADVOS represents a novel type of extracorporeal nonbiologic liver support system and holds promise to work as a bridging or destination therapy for patients with ACLF either awaiting liver transplantation or recovery of liver function. In contrast to previous albumin dialysis systems, for ADVOS only a minimum amount of human albumin (200 ml) is needed, saving this scarce resource in clinics. Beyond liver support, ADVOS corrects severe metabolic and respiratory acid-base disequilibrium, which is relevant for multi-organ failure therapy [10, 13].

In this current study, we were able to demonstrate the applicability and safety of ADVOS in a discontinuous treatment setting for patients with ACLF and the need for renal replacement therapy outside of an intensive care unit.

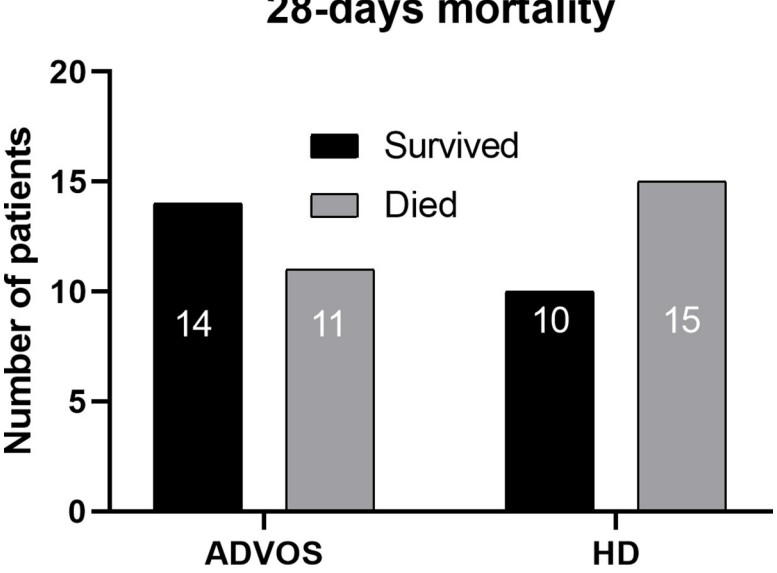

**Fig 3. 28-days mortality of patients treated with ADVOS vs. HD.**

Included patients had a severe liver dysfunction reflected by high MELD- and CLIF-C ACLF scores and liver transplantation was an option only for one patient, while for the majority allocation of donor organ was not available due to ongoing alcohol abuse. Thus, our cohort represents typical cases of ACLF, where patients cannot qualify for allocation of a donor organ and therapeutic options to support their liver function are scarce.

By now, there were only two studies published, which have tested ADVOS as continuous treatment in intensive care. The first study was published by Huber et. al. and evaluated the applicability of ADVOS for patients with ACLF or secondary liver failure. All 14 included patients were critically ill, diagnosed with HRS-AKI and the need for renal replacement therapy. The studied cohort was comparable to our current cohort with a mean MELD score of 34 (SD±7), age of 54 (SD±13) and 57% of men, being comparable to our cohort (Table 2). They found that ADVOS reduced significantly mean bilirubin (-8.3 mg/dl SD±6.5), creatinine (-0.6 mg/dl SD±0.6) and BUN (-18 mg/dl SD±16) after a mean time of dialysis of 9.6 h (SD±3.2). While our data confirm these findings that ADVOS decreases the levels of serum bilirubin, creatinine and BUN, there are still some differences between both studies. The most striking one is the reduction of bilirubin, which was more than 2-fold higher in the study by Huber et al. when compared to our data (-3.4 vs. -8.3 mg/dl). In contrast, the reduction of BUN was nearly 2-fold higher in our patients than in the study by Huber et al (-33.7 vs. -18 mg/dl, Table 3). We hypothesize that the lower reduction protein-bound bilirubin in our study could be explained by the absence of vasopressors and, thus, lower mean arterial pressure during dialysis in our cohort (S3 Table). Because dialysis efficacy also relies on sufficient mean arterial pressure, which might become more evident for protein-bound compounds for albumin dialysis [18]. Despite Huber et al. did not declare the exact number of patients who received vasopressors in their study as well as mean blood pressure during dialysis, their exclusion criteria excluded 1.) patients who received an excessive amount of vasopressor support (Dopamine >15 μg/kg/min, epinephrine >0.1 μg/kg/min or norepinephrine >0.1 μg/kg/min); 2.) patients with mean arterial pressure ≤50 mmHg despite conventional medical treatment, which would include vasopressor therapy. Therefore, it seems obvious that most of the patients received vasopressor treatment during ADVOS, which could have stabilized blood pressure and, thus, promoted a more efficient dialysis. Further, we found that elimination of bilirubin is less effective when anticoagulation is performed with citrate instead of heparin as seen in other ongoing pilot studies (Weinmann-Menke J et al., unpublished data of our group). This might have also contributed to the lower reduction of bilirubin in our cohort as we have applied citrate anticoagulation, while Huber et al. used heparin as anticoagulant during ADVOS [10].

The second study was published by Fuhrmann et. al. who evaluated ADVOS as continuous treatment for critically ill patients with multi-organ failure in intensive care [13]. They included 34 patients within 15 (44%) with ACLF and 19 (56%) with acute-liver failure (ALF). Further baseline characteristics like median age (59 years, IQR 46; 72) and percentage of male patients (68%) were similar to our cohort (Table 2). They reported a median reduction of bilirubin of -20% (IQR -34; -4), BUN -48% (IQR -62; -17) and creatinine -24% (IQR -49; -5) after a median dialysis time of 18.5 h (IQR 8.25; 22.0). Comparing these values to our study, we found that ADVOS was similarly efficient as discontinuous treatment when performed for 16 h. Discontinuous ADVOS reduced median bilirubin of -14.5% vs. -20%, BUN of -33.7% vs. -48%, creatinine of -11.8% vs. -24% (Table 3). In line with our data, they found that the reduction of bilirubin was more efficient at higher starting levels of bilirubin (S4 Table).

Beside detoxification, safety must be ensured to recommend ADVOS for discontinuous use outside of intensive care units. In our current study we did not detect any safety issues. Treatments were well tolerated, and no treatment related adverse events were documented during ADVOS. Strikingly, we observed no bleeding events despite primary and secondary

hemostasis were severely compromised (low platelet count and high prothrombin time-INR, Table 2). In contrast, Fuhrmann et al. reported three major bleeding complications during ADVOS treatment with heparin anticoagulation.

However, it must be acknowledged that our study was retrospective in design. Therefore, we were only able to evaluate documented safety issues and patients were not followed after a predefined protocol. Therefore, these findings have to be interpreted with caution and must be validated in future prospective trials.

In a preliminary survival analysis, ADVOS showed non-inferiority in 28-days morality compared a matched cohort treated with regular hemodialysis (11 vs. 15 patients survived). Additionally, the 28-days mortality of our ADVOS patients was lower as predicted by the CLIF-C ACLF score (44 vs. 56%). However, 28-days and long-term survival analysis between ADVOS and HD did not reach significance (Fig 3, p = 0.225; S1 Fig, p = 0.08) and again, these results have to be interpreted with caution due to the retrospective design of this study. Though, our findings may be a first hint that ADVOS may have a beneficial effect on survival of patients with ACLF and may be helpful for designing future randomized-controlled trials.

Our study has limitations that need to be acknowledged. First, as mentioned before this is a retrospective, case-control study and therefore the results have to be interpreted according to the study's design. Moreover, our study cohort is small and precludes further detailed analyses. Last, this study lacks a randomized control group. Despite a 1:1 matching of ADVOS and HD patients according to defined criteria, selection bias might have occurred.

In conclusion, we report for the first time the feasibility and safety of ADVOS as a discontinuous treatment in patients with ACLF outside of an intensive care unit. Additionally, we found a similar detoxification effect of ADVOS as discontinuous dialysis compared to previous data, where ADVOS was performed in a continuous dialysis setting. We found that ADVOS was not inferior in 28-days morality compared to regular hemodialysis in patients with ACLF, future randomized trials are needed to investigate its prognostic effect compared to regular HD.

## Supporting information

**S1 Fig. Long-term survival of ADVOS vs. HD treated patients (*group difference calculated by Log-rank test; the one patient who received liver transplantation was not censored in the analysis).**
(TIF)

**S1 Table. Baseline characteristics of patients of the local ACLF registry, who were used to find appropriate matches for ACLF patients who were treated with ADVOS.**
(DOCX)

**S2 Table. ADVOS treatment parameters.**
(DOCX)

**S3 Table. Safety parameter before and after the last ADVOS treatment.**
(DOCX)

**S4 Table. Concentration dependent elimination of bilirubin after cumulative 16 h of ADVOS in discontinuous treatment.**
(DOCX)

## Author Contributions

**Conceptualization:** L. Kaps, C. J. Ahlbrand, J. Weinmann-Menke.

**Data curation:** L. Kaps, C. J. Ahlbrand, M. Nagel, J. Weinmann-Menke.

**Formal analysis:** C. Labenz.

**Investigation:** L. Kaps, C. J. Ahlbrand, R. Gadban, M. Nagel, P. Klimpke, S. Holtz, M. Michel, W. M. Kremer, M. Hilscher, D. Kraus, J. M. Schattenberg.

**Methodology:** L. Kaps.

**Project administration:** L. Kaps, C. J. Ahlbrand.

**Resources:** L. Kaps, M. Nagel, P. R. Galle, D. Kraus, J. Weinmann-Menke.

**Software:** L. Kaps.

**Supervision:** J. Weinmann-Menke.

**Validation:** J. Weinmann-Menke.

**Writing – original draft:** L. Kaps, C. Labenz.

**Writing – review & editing:** C. J. Ahlbrand, P. Klimpke, S. Boedecker, J. M. Schattenberg, J. Weinmann-Menke.

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
