## [Decision Letter · Decision Letter 0]

16 Feb 2021

PONE-D-21-02242

Applicability and safety of discontinuous ADVanced Organ Support (ADVOS) in the treatment of patients with acute liver failure (ACLF) outside of intensive care.

PLOS ONE

Dear Dr. Julia Weinmann-Menke,

Thank you for submitting your manuscript to PLOS ONE. After careful consideration, we feel that it has merit but does not fully meet PLOS ONE’s publication criteria as it currently stands. Therefore, we invite you to submit a revised version of the manuscript that addresses the points raised during the review process.

We look forward to receiving your revised manuscript.

Kind regards,

Tatsuo Kanda, M.D., Ph.D.

Academic Editor

PLOS ONE

Journal Requirements:

2. Please include your tables as part of your main manuscript and remove the individual files. Please note that supplementary tables (should remain/ be uploaded) as separate "supporting information" files

4.Thank you for stating the following in the Acknowledgments Section of your manuscript:

"Supported by internal research funds of the University Medical Centre Mainz"

Reviewers' comments:

Reviewer's Responses to Questions

**Comments to the Author**

1. Is the manuscript technically sound, and do the data support the conclusions?

Reviewer #1: Yes

Reviewer #2: Yes

2. Has the statistical analysis been performed appropriately and rigorously? 

Reviewer #1: Yes

Reviewer #2: Yes

3. Have the authors made all data underlying the findings in their manuscript fully available?

Reviewer #1: Yes

Reviewer #2: Yes

4. Is the manuscript presented in an intelligible fashion and written in standard English?

Reviewer #1: Yes

Reviewer #2: Yes

5. Review Comments to the Author

Reviewer #1: Authors concluded discontinuous ADVOS treatment outperformed hemodialysis in reducing protein-bound metabolites. The data of the article is attractive. However, authors need to be more careful not to mislead. Revisions required are as follows:

Major revisions

#1: There is no significant difference between 28-days survival in ADVOS treated cohort and HD treated cohort. Avoid misleading expressions and keep the content that it was possible to indicate non-inferiority.

#2: Concerning 44% (IQR 30; 85), is 28-days ‘survival’ correct or 28-days ‘mortality’ correct? The description in the text and the description in the table do not match.

#3: It would be better if there is a comparison between predicted 28-days survival in the ADVOS treated cohort and HD treated cohort.

Minor revisions

#1: In Abstract, in line 19, the correct one is ‘‘stronger’’, not ‘‘stringer’’.

#2: In Results, in line 2, the correct one is ‘‘65.3%’’, not ‘‘53.5%’’.

#3: Concerning main etiology of cirrhosis in three patients, is ‘a mixed etiology (HBV, alcohol or NASH)’ correct or ‘viral hepatitis (HBV, HCV)’ correct? The description in the text and the description in the table do not match.

#4: In Results, in line 8, what is ‘hydropy’?

#5: In Results, in line 12, and In Table 2 and In Supplementary table 3, the correct one is ‘‘prothrombin time-INR’’, not ‘‘INR’’.

Reviewer #2: To evaluate the effect of ADVOS as discontinuous treatment for patients with ACLF outside intensive care unit, Leonard Kaps et al. conducted retrospective study including 26 patients with ACLF and HRS-AKI. Comparing with hemodialysis performed in a matched cohort, ADVOS achieved a stringer decrease in bilirubin (p=0.01), while detoxification of watersoluble catabolites’ including creatinine and BUN was comparable.

They concluded that discontinuous ADVOS treatment was safe and effective in patients with ACLF outside intensive care and outperformed hemodialysis in reducing protein-bound metabolites.

This is an interesting and valuable paper reporting on the effect of discontinuous ADVOS treatment for patients with ACLF and HRS-AKI. Although the paper is well written and clearly presented, the authors should reconsider the following issues.

Major comments:

1. The authors described in the summary that ‘The 28-days survival rate in the ADVOS group was 56% (14/26) exceeding the expected survival time according to 12% (predicted median 28-days survival was 44%, IQR 30; 85)’ and in the conclusion section ‘we found a trend that ADVOS carries a survival advantage in patients with ACLF.’

These sentences seem to be sophistical argument. Predicted 28-days survival was calculated in each patient by the CLIF-C ACLF calculator. It should not be described that the 28-days survival rate in the ADVOS group exceeded the expected value by 12%.

The effect of ADVOS on the 28-days survival of patients with ACLF should be assessed statistically according to the result of case control study: 56% (14 of 25) in the ADVOS group vs 40% (10 of 25) in the HD group. Log-rank test is also able to be used for survival analysis.

2. In Table 2, the data of ACLF grade should be shown in numbers of patients belonging to each grade: grade 1, 2, 3, instead of median and IQR.

3. Long-term outcome of 26 patients with ACLF was not disclosed. It should be shown whether any patient received liver transplantation or not.

4. Urine volume could be assessed before and after treatment with ADVOS and with HD.

5. The title should be corrected partially: patients with acute liver failure (ACLF)⇒patients with acute-on-chronic liver failure (ACLF).

6. The authors should ensure that tables are the best format to present their data.

7. The description of matching criteria 1 in Table 4 is insufficient. How many patients were male?

6. PLOS authors have the option to publish the peer review history of their article (what does this mean?). If published, this will include your full peer review and any attached files.

Reviewer #1: **Yes: **Hidehiro Kamezaki

Reviewer #2: **Yes: **Nobuaki Nakayama

---

## [Author Response · Author response to Decision Letter 0]

22 Feb 2021

Dear Dr. Kanda, 

We would like to thank you for returning the reviewers and editors comments and providing us with the opportunity to resubmit after major revisions. We found the suggestions very helpful and incorporated the raised points accordingly. The revised manuscript contains all changes highlighted and a point-to-point response is included below. We believe that the comments and consecutive changes improved the manuscript and would hope that you can accept the manuscript for publication in PLOS ONE.

Sincerely yours, 

Julia Weinmann-Menke, MD

---

## [Decision Letter · Decision Letter 1]

8 Mar 2021

PONE-D-21-02242R1

Applicability and safety of discontinuous ADVanced Organ Support (ADVOS) in the treatment of patients with acute liver failure (ACLF) outside of intensive care.

PLOS ONE

Dear Dr. Julia Weinmann-Menke,

Thank you for submitting your manuscript to PLOS ONE. After careful consideration, we feel that it has merit but does not fully meet PLOS ONE’s publication criteria as it currently stands. Therefore, we invite you to submit a revised version of the manuscript that addresses the points raised during the review process.

We look forward to receiving your revised manuscript.

Kind regards,

Tatsuo Kanda, M.D., Ph.D.

Academic Editor

PLOS ONE

Journal Requirements:

Reviewers' comments:

Reviewer's Responses to Questions

**Comments to the Author**

1. If the authors have adequately addressed your comments raised in a previous round of review and you feel that this manuscript is now acceptable for publication, you may indicate that here to bypass the “Comments to the Author” section, enter your conflict of interest statement in the “Confidential to Editor” section, and submit your "Accept" recommendation.

Reviewer #1: All comments have been addressed

Reviewer #2: All comments have been addressed

2. Is the manuscript technically sound, and do the data support the conclusions?

Reviewer #1: Yes

Reviewer #2: Yes

3. Has the statistical analysis been performed appropriately and rigorously? 

Reviewer #1: Yes

Reviewer #2: Yes

4. Have the authors made all data underlying the findings in their manuscript fully available?

Reviewer #1: Yes

Reviewer #2: Yes

5. Is the manuscript presented in an intelligible fashion and written in standard English?

Reviewer #1: Yes

Reviewer #2: Yes

6. Review Comments to the Author

Reviewer #1: (No Response)

Reviewer #2: Leonard Kaps et al. revised their manuscript carefully and adequately.

They disclosed that only one patient in the ADVOS cohort received liver transplantation and created supplementary figure1.

They should describe whether a liver-transplanted patient was censored at the time of transplantation in the long-term survival analysis or not (Log-rank test, supplementary figure 1, p=0.08).

7. PLOS authors have the option to publish the peer review history of their article (what does this mean?). If published, this will include your full peer review and any attached files.

Reviewer #1: **Yes: **Hidehiro Kamezaki

Reviewer #2: **Yes: **NOBUAKI NAKAYAMA

---

## [Author Response · Author response to Decision Letter 1]

9 Mar 2021

We apologize that this detail escaped our attention and agree that it should be noted whether the transplanted was censored or not in the analysis. We treated the transplanted patient as a complete case and, thus, he was not censored. Accordingly, we added the following passage in the methods section “The one patient was transplanted due to terminal hepatic failure and was not censored in the analysis, being treated as a complete case”. Further, we mentioned in the description of supplementary figure 1 “the one patient who received liver transplantation was not censored in the analysis”.

---

## [Editor Report · Decision Letter 2]

17 Mar 2021

Applicability and safety of discontinuous ADVanced Organ Support (ADVOS) in the treatment of patients with acute-on-chronic liver failure (ACLF) outside of intensive care.

PONE-D-21-02242R2

Dear Dr. Julia Weinmann-Menke,

We’re pleased to inform you that your manuscript has been judged scientifically suitable for publication and will be formally accepted for publication once it meets all outstanding technical requirements.

Kind regards,

Tatsuo Kanda, M.D., Ph.D.

Academic Editor

PLOS ONE
---

## [Editor Report · Acceptance letter]

23 Mar 2021

PONE-D-21-02242R2 

Applicability and safety of discontinuous ADVanced Organ Support (ADVOS) in the treatment of patients with acute-on-chronic liver failure (ACLF) outside of intensive care. 

Dear Dr. Weinmann-Menke:

I'm pleased to inform you that your manuscript has been deemed suitable for publication in PLOS ONE. Congratulations! Your manuscript is now with our production department. 

Kind regards, 

on behalf of

Dr. Tatsuo Kanda 

Academic Editor

PLOS ONE